Laser communications system with drones as relay medium for healthcare applications

Sait Adeeb adeebzeus@gmail.com 1
Al-Hadhrami Tawfik tawfik.al-hadhrami@ntu.ac.uk 1
Saeed Faisal 2
Basurra Shadi 2
Qasem Sultan Noman 3
1 Computer Science Department, School of Science and Technology, Nottingham Trent University , Nottingham , United Kingdom
2 DAAI Research Group, College of Computing and Digital Technology, Birmingham City University , Birmingham , United Kingdom
3 Computer Science Department, College of Computer and Information Sciences, Imam Mohammad Ibn Saud Islamic University (IMSIU) , Riyadh , Saudi Arabia
Sohaib Osama
Electronic publication date: 2024 Feb 7
Publication date: 2024
Volume: 10
Electronic Location ID: e1759
Received 2023 Aug 17; Accepted 2023 Nov 26
Copyright: ©2024 Sait et al.
Copyright year: 2024
Copyright holder: Sait et al.
License: This is an open access article distributed under the terms of the Creative Commons Attribution License, which permits unrestricted use, distribution, reproduction and adaptation in any medium and for any purpose provided that it is properly attributed. For attribution, the original author(s), title, publication source (PeerJ Computer Science) and either DOI or URL of the article must be cited.
License URL: https://creativecommons.org/licenses/by/4.0/

Keywords: Laser communications, UAV-drones, Remote communications, Healthcare advance communications

Funding: Imam Mohammad Ibn Saud Islamic University (IMSIU) RP-21-07-09 This work was funded by the Deanship of Scientific Research at Imam Mohammad Ibn Saud Islamic University (IMSIU) through Research Partnership Program no RP-21-07-09. The funders had no role in study design, data collection and analysis, decision to publish, or preparation of the manuscript.

==============================
This article introduces a prototype laser communication system integrated with uncrewed aerial vehicles (UAVs), aimed at enhancing data connectivity in remote healthcare applications. Traditional radio frequency systems are limited by their range and reliability, particularly in challenging environments. By leveraging UAVs as relay points, the proposed system seeks to address these limitations, offering a novel solution for real-time, high-speed data transmission. The system has been empirically tested, showcasing its ability to maintain data transmission integrity under various conditions. Results indicate a substantial improvement in connectivity, with high data transmission success rate (DTSR) scores, even amidst environmental disturbances. This study underscores the system’s potential for critical applications such as emergency response, public health monitoring, and extending services to remote or underserved areas.

Introduction

Advancements in communication technology over the past decades have been pivotal in enhancing human connectivity across vast distances. Yet, despite the progress, industries ranging from disaster response to healthcare still face challenges in establishing reliable high-speed communications in remote areas. Traditional radio frequency (RF) systems often fall short due to signal attenuation, interference, and limited bandwidth, especially in geographically isolated regions (Stojanovic & Preisig, 2009).

Laser communication systems, known for their broad bandwidth and low latency, have emerged as a promising solution to these challenges (Boroson & Robinson, 2014). Conceptualised in the 1960s and recently exemplified by the Lunar Laser Communication Demonstration (LLCD) (Boroson & Robinson, 2014), these systems show potential but are hindered by the necessity of direct line-of-sight and substantial infrastructure (Schawlow & Townes, 1958).

To navigate these limitations, this article explores the integration of laser communication with drone technology. Drones offer a dynamic and versatile platform to establish direct communication links, bypassing the line-of-sight constraints and infrastructure dependencies (Ranjha, Kaddoum & Dev, 2022). They have the potential to deliver rapid, real-time data transmission in critical areas, such as during disaster relief efforts and within healthcare facilities, to ensure the accuracy and timeliness of patient data management.

Despite the progress, there remains a gap in deploying these systems for practical healthcare applications in remote settings. Previous research has largely focused on the technical development of laser communication and drones independently, with less emphasis on their combined application in dynamic, real-world environments. Figure 1 shows an example of how the data can be transmitted to the drone to enable communication in remote areas where direct communication is difficult.

Addressing this gap, our methodology involves the development and testing of a UAV-based laser communication prototype designed to operate under various environmental conditions. This research contributes to the field by:

1. Demonstrating the feasibility of using UAVs as reliable relay mediums for laser communication in remote areas.

2. Evaluating the system’s performance against traditional RF systems, particularly within indoor environments.

3. Presenting potential applications for emergency response and healthcare, highlighting the system’s capacity to support real-time critical data transmission.

The remainder of this article is organised as follows: ‘Related Work’ reviews existing work in the field, ‘Methodology: Development and Evaluation of a Drone-Enhanced Laser Communication System’ outlines the methodology and system design, ‘Results and Discussion’ presents the results and analysis, and ‘Conclusion’ concludes the article with a summary of findings and potential future work.

Related Work

Radio frequency communication systems: traditional and limiting factors

Traditional radio frequency (RF) communication systems have long been the foundation of wireless communication. They utilise radio waves to transmit information, differing from laser communication’s use of light waves. While RF systems facilitate various wireless applications, they face limitations in frequency range, modulation complexity, line-of-sight requirements, susceptibility to interference, security vulnerabilities, and high power consumption, particularly in remote and inaccessible regions (Fette, 2008).

Figure 1 UAV trajectory planning.

Advancements in laser communication systems

Laser communication systems, or free-space optical (FSO) communication, offer significant advantages over RF systems in terms of data rates and beamwidth, operating within the near-infrared wavelength region to optimize transmission through atmospheric windows (Sibley, 2020). Recent advancements have focused on increasing the capacity and reliability of these systems, addressing challenges in modulation and spectrum regulation (Giggenbach & Shrestha, 2022).

Comparing recent works, such as Hassan (2022), which explored ground-based FSO systems for urban environments, highlights the necessity for expanded reach in remote areas (Jaafar & Yanikomeroglu, 2021). These studies reveal a gap in current FSO applications: the lack of mobile, flexible systems that can adapt to varying topographies and infrastructural limitations, especially in healthcare contexts.

Innovative integration of laser communication systems with UAV drones

UAV drones offer a flexible platform for the integration of laser communication systems (Fawaz, Abou-Rjeily & Assi, 2018). However, its integration faces several obstacles, including the requirement for communication modules that are lightweight and compact, limitations on available power, and the possibility that dynamic flight conditions may interfere with pointing, acquisition, and tracking (PAT) systems (Alzenad et al., 2016).

The problem in focus is the need for a communication system that transcends the limitations of traditional RF systems, particularly in providing stable and reliable communication for healthcare services in remote areas. There is a pressing need for innovative approaches that can ensure continuous and high-speed communication, which is vital for remote diagnostics, telemedicine, and emergency medical response.

Several studies have concentrated on the research and development of laser communication systems that are based on drones. For instance, one study proposed an airborne free space optical (FSO) network that made use of drones as relay nodes, with the goal of optimizing the network structure to maximize throughput and coverage (Hassan, 2022). Drones can be equipped with laser communication modules and strategically positioned to establish and maintain line-of-sight (LOS) between transmitter and receiver nodes (Wu, Sun & Ansari, 2020). Drones can help overcome obstacles that may obstruct the line of sight, such as buildings or terrain by offering a high degree of mobility and the ability to adapt to changing environmental conditions. Additionally, drones can be used to establish a temporary communication network in areas where the implementation of permanent infrastructure may be difficult or has failed, such as in remote or disaster-stricken areas (Khosravi et al., 2021).

A fleet of drones can be used to form a network of relay nodes, which creates the way for the establishment of a laser communication system over wide distances or in locations with several obstacles (Fawaz, Abou-Rjeily & Assi, 2018). The relay system will be able to adapt to changing conditions due to the utilization of swarm intelligence and the cooperative behaviour of a fleet of drones. This helps to guarantee that the line of sight is preserved along the entire communication channel.

The rapid deployment of a fleet of drones can support the quick formation of a communications network, which is particularly beneficial in emergency response situations or in areas where traditional communication networks are unavailable or insufficient (Chen et al., 2015). Moreover, a fleet of drones would provide stability and fault tolerance in the communication network (Mohammadnia et al., 2021). This ensures that data transmission will continue without interruption even if one or more drones in the fleet have technical difficulties or are unable to keep line of sight.

The laser power beaming technology involves the transmission of energy from a laser source located on the ground to a drone mounted with photovoltaic cells, which converts the energy from the laser beam into electrical energy (Chen et al., 2015a). This method supplies the drone with continuous power, which enables it to remain airborne for significantly longer periods without the need for the batteries to be either replaced or recharged.

Drones can overcome the restrictions imposed by typical battery-based power sources by utilizing laser power beaming. This allows for longer flight periods, higher cargo capacity, and better operational flexibility (Mohammadnia et al., 2021).

The laser communication system that uses drones as relays has significant potential to improve healthcare systems, especially in remote and underserved areas. One important use is in telemedicine, where the laser communication system’s high bandwidth and low latency can make it easy to send high-resolution medical images, video consultations, and real-time monitoring of patient data from remote healthcare facilities to specialized medical centres (Hayat, Yanmaz & Muzaffar, 2016). This can bridge the gap between healthcare services in rural areas and those in cities, giving people in remote areas access to specialized care and expertise without having to travel far.

Also, the mobility and flexibility of drones can be used to quickly transport medical supplies such as vaccines, medications, and blood samples, along with real-time communication and tracking through a laser communication system. This ability is especially helpful in emergencies, disaster response, and pandemics, where quick delivery of medical aid and good communication between medical professionals are key to saving lives and lessening the effects of crises (Juneja, 2021).

The proposed system could also facilitate the seamless collection and transmission of critical health data from remote areas, enabling effective tracking and assessment of public health trends and potential risks. This enhanced data flow has the potential to empower healthcare authorities with informed decision-making, optimized resource allocation, and targeted interventions to address the specific health requirements of remote populations. By combining laser communication systems with drones, the proposed solution can greatly improve connectivity and data transfer in the health sector. This can open up new ways to improve healthcare delivery, reduce differences in access to medical services, and help people in remote and underserved areas achieve better health outcomes.

Emerging studies in UAV communication highlight the potential of integrating deep reinforcement learning (DRL) to optimize system performance in complex networking environments. For instance, recent advancements have explored the synergy between UAV and reconfigurable intelligent surface (RIS) technologies, positioning them as pivotal for future wireless networks. Such studies underscore the viability of DRL as a technique for enhancing the adaptability and efficiency of UAV-assisted communication systems, which is essential for dynamic and resource-constrained settings typical in remote healthcare applications (Nguyen, Park & Park, 2023).

Multi-drone network

A fleet of drones can be used to form a network of relay nodes, which creates the way for the establishment of a laser communication system over wide distances or in locations with several obstacles (Alzenad et al., 2016). The relay system will be able to adapt to changing conditions due to the utilisation of swarm intelligence and the cooperative behaviour of a fleet of drones. This helps to guarantee that the line of sight is preserved along the whole communication channel.

The rapid deployment of a fleet of drones can support the quick formation of a communications network, which is particularly beneficial in emergency response situations or in areas where traditional communication networks are unavailable or insufficient (Mahdi AJ & Ucan, 2022). Moreover, a fleet of drones would provide stability and fault tolerance in the communication network (Khosravi et al., 2021). This ensures that data transmission will continue without interruption even if one or more drones in the fleet have technical difficulties or are unable to keep line of sight. Due to laser communication systems having such a narrow beamwidth, they require extremely precise pointing, acquisition, and tracking (PAT) capabilities to set up and keep a steady communication link (Kaushal & Kaddoum, 2017). This can be a particularly difficult problem to solve for mobile platforms, such as drones or satellites because the relative motion between the transmitter and receiver must be continuously corrected to maintain a solid connection. The overall system design and implementation issues are made more difficult by the fact that PAT systems can be complicated and expensive to set up (Baister & Gatenby, 1994).

Laser communication systems in healthcare: emergency and telemedicine applications

The ability to establish and maintain effective communication systems is critical in emergencies for enabling rapid, coordinated responses and ensuring the timely delivery of life-saving medical care and supplies. Inadequate infrastructure, a lack of available resources, and unstable communication technologies are frequently the root causes of these difficulties. The demand for medical care services intensifies in the aftermath of natural disasters or military conflicts, and at the same time, the need grows for efficient and dependable communication networks.

A quickly deployable communication system, such as the laser communication systems integrated with drones, can significantly enhance healthcare access and emergency response capabilities. Such systems can support real-time coordination among, for example, first responders, medical personnel, and disaster management teams, which allows for the efficient allocation of resources and the deployment of life-saving measures. Additionally, these communication systems can facilitate the delivery of remote healthcare, health monitoring, and research initiatives, which ultimately contribute to improved health outcomes and reduced health inequalities.

Drones fitted with laser communication modules can rapidly establish communication links in places affected by a natural disaster, enabling first responders, medical professionals, and other stakeholders to coordinate their efforts in real-time (Spranger et al., 2016). These communication links have the potential to make the effective allocation of resources easier, to provide a better understanding of the situation, and to support telemedicine consultations for emergency medical care (Gupta, Jain & Vaszkun, 2016).

Indoor drones equipped with laser communication modules can enhance healthcare provision in indoor environments such as hospitals and healthcare homes. These drones can navigate efficiently within these spaces and facilitate the real-time transmission of critical patient data, supporting patient monitoring and immediate responses to patient needs (Gupta, Jain & Vaszkun, 2016). This approach also avoids potential interference and health concerns associated with radio frequency signals, common in traditional communication systems (Spranger et al., 2016). The use of infrared lasers ensures reliable, high-speed data transmission across varying indoor lighting conditions, improving patient monitoring accuracy (Sangaiah & Mukhopadhyay, 2020). This innovation reduces healthcare professionals’ workload, thereby facilitating more efficient and effective patient-centric care.

Methodology: development and evaluation of a drone-enhanced laser communication system

This section articulates the comprehensive research procedure for the development and evaluation of a prototype laser communication system (LCS). This system, which is unique in its application, employs drones as dynamic relay nodes for the transmission and reception of information via laser signals. The focus of this methodology is to assess the system’s efficacy, dependability, and practicality in a controlled setting.

The proposed laser communication system possesses a distinctive edge over existing systems through its use of drones as adaptable relay nodes. This feature enables the system to mitigate one of the inherent limitations of traditional laser communication systems, the requirement for an unobstructed line of sight between transmitter and receiver. In this way, the system enhances connectivity under challenging circumstances. Additionally, the implementation of a drone fleet introduces a swift, scalable, and flexible communication infrastructure, broadening the system’s potential applications across numerous sectors.

This research methodology emphasizes the development and evaluation of a prototype system, pivotal to proving the system’s feasibility and functionality. The creation of the prototype validates the proposed system’s fundamental concepts, design choices, and execution strategies, underlining the system’s potential and efficacy. The prototype also offers valuable insights into the system’s performance across an array of scenarios.

The prototype system is a tangible demonstration of the laser communication system’s ability to address its intended objectives. These include surpassing the restrictions of conventional radio frequency-based systems and augmenting connectivity in remote or difficult-to-reach areas. The research methodology facilitates the identification of potential shortcomings, and possible improvements, and demonstrates potential applications in a variety of sectors, notably, emergency healthcare, through the testing and development of a fully functional prototype system.

The proposed laser communication system as shown in Fig. 2, is an integration of drones equipped with laser communication modules and a ground control station. The objective of the system is to transmit and receive data in the form of laser signals via drones serving as relay nodes, thus ensuring an unobstructed line of sight necessary for laser communication.

The system’s primary components are:

• Drone: The drone is fitted with a laser communication module which comprises a laser transmitter and a photodiode receiver. The drone functions as a relay node, receiving data from the ground station and transmitting it to the destination. The drone’s mobility ensures an uninterrupted line of sight, a fundamental requirement for successful laser communication.

• Laser communication module: This module is responsible for converting data into laser signals for transmission and converting received laser signals back into data. It is composed of a laser diode that acts as the transmitter and a photodiode that functions as the receiver.

• Ground control station: This is the starting point of communication where data is prepared for transmission. The ground control station converts the data into laser signals and transmits them to the drone. It also serves to receive data transmitted by drones.

The operation of the system begins at the ground control station where data is converted into laser signals. These signals are then transmitted to the drone, which serves as the relay medium. The drone receives these laser signals using its photodiode receiver. Upon receiving the data, the drone relays the data to the next node or the destination by transmitting the data as laser signals from its laser diode. This process repeats until the data reaches its destination.

Figure 2 Proposed system design.

Throughout this process, the drones maintain a clear line of sight, ensuring the laser signals are not interrupted. This configuration enables the system to overcome the inherent limitations of traditional laser communication systems, providing enhanced and reliable communication links in diverse environments.

Several scenarios and situations are considered that aim to test the efficiency, reliability, and robustness of the laser communication system. The system’s performance under various conditions, including different lighting conditions, movement of the endpoints, and the ability to maintain stable communication links will be thoroughly tested. Through these scenarios and discussions, we aim to gain a deeper understanding of the system’s capabilities, limitations, and potential for further development. By identifying areas of improvement and addressing challenges, we can refine the prototype and increase its effectiveness, ultimately enabling a more robust and efficient communication system for the healthcare sector and other applications.

LCS prototype architecture

LCS consists of several parts and these components combined work simultaneously to provide a suitable prototype system that can be used to carry out research on various properties of using laser communication with drones as shown in Fig. 3. Three different components contribute to the functionality of the LCS:

• Laser communication system: Serial communication will be set up between the PCs and the Arduinos to interpret and send data directly. The laser communications system acts as the wireless mode of communication between the computers for the transmission of different types of data.

• Actuators for controlling the pan and tilt movement of the pointing and tracking system: Two servo motors that can control the motion of the laser/solar cell on the x and y axis to enable efficient tracking of the drone are required on each side.

• Adapting drone for GPS and IMU data: Since the drone used for this prototype, the DJI Tello, does not have a built-in GPS module, an additional GPS module called TTGO T-Beam V1.1 needs to be attached to the drone to receive continuous readings of the drone’s position.

Dynamic identification and tracking

The relay drone is equipped with an advanced sensor suite, including GPS, and inertial measurement units (IMUs), enabling it to accurately identify and maintain alignment with the transmitter and receiver units. Sophisticated tracking algorithms utilise these sensors to compensate for environmental dynamics, such as changes in position or obstacles, ensuring a stable line-of-sight is maintained. These algorithms are critical for the system’s adaptability, allowing for real-time adjustments to the drone’s flight path to preserve the integrity of the communication link.

Figure 3 LCS prototype architecture.

LCS prototype components

Laser transmitter and receiver algorithms

The transmitter algorithm ( Algorithm 1 ) is responsible for providing a menu-based interface for the user to select between transmitting text, audio, or image data. The code initializes the appropriate pins and sets up serial communication as shown in the algorithm below. The user input is read, and based on the selection, the corresponding data is transmitted as ASCII values through the laser. For text transmission, the user is prompted to enter the text, which is then transmitted. For audio transmission, 10 s of audio input is collected and transmitted. For image transmission, an image captured from the camera is transmitted.

The receiver algorithm ( Algorithm 2 ) is responsible for detecting the incoming data from the laser and identifying the type of data being transmitted (text, audio, or image) as shown in the algorithm below. The code initializes the appropriate pins and sets up serial communication. It continuously reads the solar cell input and checks for a start bit. Based on the detected data type, the corresponding data is received and processed. For text reception, the text is received and printed on the serial monitor. For audio reception, 10 s of audio data are received and played through the speaker. For image reception, the image data is received and displayed on the screen.

In the LCS, the transmitter algorithm ( Algorithm 1 ) orchestrates the conversion of data types—text, audio, and image—into laser signals. It is a key interface where users command the system to encode information into modulated light for transmission. The receiver algorithm ( Algorithm 2 ) complements this by decoding the laser-modulated signals back into their original formats. These algorithms ensure seamless data exchange, translating user inputs into communication actions.

The drone’s role is pivotal; it acts as a dynamic reflector and relay point. It maintains the necessary line of sight for the laser beam, which is crucial due to the directional nature of laser communication. The drone’s onboard systems, directed by the GPS and tracking algorithms ( Algorithm 3 ), adjust its position to preserve a stable transmission path, compensating for any movement or environmental changes that could disrupt the signal. This dynamic positioning is crucial for the LCS’s ability to provide reliable communication in varied terrains and conditions.

Furthermore, the GPS algorithm ( Algorithm 4 ) is integral to the system’s spatial awareness, allowing precise geolocation tracking. It enables the laser pointer to adjust in real-time, directing the laser beam accurately towards the moving drone. The combination of these algorithms constitutes a robust framework for the LCS, facilitating a resilient communication link that is essential for critical applications, such as in remote healthcare delivery.

Algorithm 1. Laser Transmitter for LCS:	
TransmitterAlgorithm: Designed to interface with the user, allowing them to select the type of data for transmission. It encodes textual, audio, or image data into a laser-readable format, turning on and off the laser to transmit data as a series of light pulses.	
Input: Text, Image or Audio	
Output: Laser turns on and off to transmit the converted bits of data	
1: define transmitText(message) function	
2: trim newline characters from message	
3: transmit special identifier for text data using transmitChar((char)1)	
4: for i = 0 to message.length()-1 do	
5: transmit message[i] using transmitChar(message[i])	
6: end for	
7: transmit newline character using transmitChar(’\n’)	
8: end function	
9: define transmitAudio() function	
10: transmit special identifier for audio data using transmitChar((char)2)	
11: initialize startTime as the current time in milliseconds	
12: while the difference between the current time and startTime is less than 10000 ms (10 s) do	
13: audioValue <- analogRead(audioInPin)	
14: scaledValue <- map audioValue from range 0-1023 to 0-255	
15: write scaledValue to ledPin	
16: delay for 1000000 / sampleRate microseconds	

17: end while	
18: end function	
19: define transmitImage() function	
20: transmit special identifier for image data using transmitChar((char)3)	
21: for i = 0 to 239 (representing each line of the image) do	
22: for j = 0 to 319 (representing each pixel in the line) do	
23: pixelValue <- analogRead(cameraPin)	
24: scaledValue <- map pixelValue from range 0-1023 to 0-255	
25: write scaledValue to ledPin	
26: delay for 1000000 / sampleRate microseconds	
27: end for (inner loop for j)	
28: end for (outer loop for i)	
29: end function	

Algorithm 2. Laser Receiver for LCS:	
Receiver Algorithm: Deciphers the laser pulses back into the original data types. It uses a solar cell to detect the signal and processes the light variations to reconstruct the transmitted information.	
Input: Laser signals read through the solar cell	
Output: Output the interpreted data appropriately	
1: initialize receivedMessage as an empty string	
2: initialize receivedChar as a character	
3: while true do	
4: receivedChar <- receiveChar() function	
5: if receivedChar is a newline character then break the loop	
6: receivedMessage <- receivedMessage + receivedChar	
7: if receivedMessage starts with a text identifier then do	
8: print “Received text: ” + receivedMessage	
9: end if	
10: else if receivedMessage starts with an audio identifier then do	
11: initialize startTime as the current time in milliseconds	
12: while the difference between the current time and startTime is less than 10000 ms (10 s) do	
13: audioValue <- receiveChar()	
14: scaledValue <- map audioValue from range 0-255 to 0-1023	
15: write scaledValue to audioOutPin	
16: write 0 to audioOutPin to turn off the speaker	
17: end while	
18: end else if	
19: else if receivedMessage starts with an image identifier then do	
20: for i = 0 to 239 (representing each line of the image) do	
21: for j = 0 to 319 (representing each pixel in the line) do	
22: pixelValue <- receiveChar()	

23: scaledValue <- map pixelValue from range 0-255 to 0-1023	
24: draw a pixel at position (j, i) on the screen with scaledValue	
25: end for	
26: end for	
27: end else if	

Drone tracking, pointing, and GPS algorithms

The combined system consists of a Python script running on the master node and an Arduino sketch running on a T-Beam. The Python script connects the DJI Tello drone and reads its inertial measurement unit (IMU) data for orientation values. It also sets up a socket server to communicate with the T-Beam. The T-Beam continuously reads Global Positioning System (GPS) coordinates from its built-in GPS module and sends them over Wi-Fi to the Python script. Upon receiving GPS coordinates, the Python script that follows Algorithm 3 , calculates the pan and tilt angles required to point the transmitter/receiver toward the drone. The servos controlling the pan and tilt are then adjusted accordingly.

Algorithm 3. Moving the Laser Pointer towards the drone:	
Drone Tracking and Pointer Algorithm: Ensures the LCS’s laser pointer dynamically tracks the drone’s position, adjusting the pan and tilt mechanism in real-time based on GPS and IMU data, maintaining a consistent line of sight.	
Input: GPS and IMU values from the drone	
Output: Pan and Tilt motors move to the required angles	
1: initialize R as 6371000 (Earth’s radius in meters)	
2: initialize ground_station_lat as 12.971598 (replace with ground station’s latitude)	
3: initialize ground_station_lon as 77.594566 (replace with ground station’s longitude)	
4: define haversine_distance(lat1, lon1, lat2, lon2) function	
5: convert lat1, lon1, lat2, lon2 to radians	
6: calculate dlat as the difference between lat2 and lat1	
7: calculate dlon as the difference between lon2 and lon1	
8: calculate a using the haversine formula	
9: calculate c as 2 * atan2(sqrt(a), sqrt(1 - a))	
10: return the product of R and c	
11: define calculate_angles(latitude, longitude) function	
12: calculate distance using the haversine_distance function	
13: calculate dlon as the difference between longitude and ground_station_lon	
14: calculate pan_angle using the formula for calculating the bearing	
15: convert pan_angle to degrees	
16: set tilt_angle as 0	
17: return pan_angle and tilt_angle	
18: define handle_client(client_socket) function	
19: while true do	

20: decode the data received from the client_socket	
21: if no data received, break the loop	
22: get lat and lon converting to float	
23: calculate pan_angle and tilt_angle, calculate_angles function	
24: set the servo position for pan and tilt angles	
25: print latitude, longitude, pan angle, and tilt angle	
26: create a Tello object	
27: connect to the Tello drone	
28: start streaming video from the drone	
29: create a server socket	
30: bind the server to all interfaces and port 12345	
31: set the server to listen for connections	
32: print “Server listening for connections...”	
33: while true do	
34: accept a new client connection	
35: print the client address	
36: call handle_client function with the client socket	
37: close the client socket	
38: end while	

Algorithm 4. Sending GPS values to Laser Pointer from drone:	
GPS Algorithm: Transmits the drone’s GPS coordinates to the laser pointer, ensuring precise aiming and optimal communication path.	
Input: Movements of the drone	
Output: GPS values sent to the Raspberry Pi	
1: define setup() function	
2: begin Serial communication at 115200 baud rate	
3: initialize GPS using gpsSerial.begin function with GPSBaud, SERIAL_8N1, pins 12 and 15	
4: start Wi-Fi connection with WiFi.begin function using ssid and password	
5: while Wi-Fi is not connected do	
6: delay for 1000 ms	
7: print “Connecting to Wi-Fi...” to the serial monitor	
8: end while	
9: print “Connected to Wi-Fi” to the serial monitor	
10: end function	
11: define loop() function	
12: while gpsSerial has available data do	
13: encode gpsSerial data using gps.encode function	
14: if gps location is updated then do	
15: create a WiFiClient object named client	
16: if client is connected to the server at server_ip and server_port then do	
17: define data as the string representation of gps location’s latitude and longitude, separated by a comma	
18: send data to the server using client.println function	

19: flush client to ensure all outgoing characters are sent	
20: end if	
21: stop the client	
22: end if	
23: end while	
24: delay for 1000 ms	
25: end function	

Laser signal tracking algorithm

The proposed LCS signal tracking algorithm utilises the drone’s GPS and IMU data to determine the exact pan and tilt angles required for the laser pointing system, thereby ensuring an accurate and efficient data transmission path. The algorithm initiates by acquiring the latitude, longitude, and altitude from the drone’s GPS module and the roll, pitch, and yaw data from the IMU. These data are converted into 3D Cartesian coordinates and relative position vectors, which are then transformed into a local East-North-Up coordinate system. With these precise coordinates, the algorithm calculates the pan and tilt angles needed for the laser to accurately target the drone. The benefit of this algorithm is the increased precision and reliability of the laser communication system, enabling it to maintain a clear line of sight and adapt to dynamic changes in the drone’s position. This enhancement directly contributes to the system’s robustness, ensuring a reliable and uninterrupted data transmission link, even in challenging environments.

The GPS Algorithm ( Algorithm 4 ) enables precise location tracking, which is essential for maintaining the line of sight between the drone and ground stations, especially in dynamic environments.

In the proposed signal tracking algorithm ( Algorithm 5 ), the GPS and IMU data from the drone is used to calculate the pan and tilt angles for the laser pointing system as shown in the algorithm below.

Algorithm 5. PAT System:	
Pointing and Tracking Algorithm : Takes GPS and IMU data to calculate the required pan and tilt angles, facilitating a direct laser communication pathway between the drone and the ground station.	
Input: Drone GPS data, Drone IMU data	
Output: Pan & tilt angles for the laser pointing system	
1. // Acquire GPS and IMU data from the drone a. DroneGPSData ← GetDroneGPSData()	
b. DroneIMUData ← GetDroneIMUData()	
2. // Find relative position between drone, transmitter & receiver	
a. X=N+h∗coscoslatitude∗coscoslongitude	
b. Y=N+h∗coscoslatitude∗sinsinlongitude	
c. Z=1−e2∗N+h∗sinsinlatitude	
Where: N=a1–e2∗sin2latitude	

a= Earth’s equatorial radius (≈ 6,378,137m)	
e= Earth’s eccentricity (approx. 0.08181919084)	
h= altitude above the ellipsoid	
3. ECEF_Coordinates ← ConvertGPStoECEF(DroneGP 4. // Calculate relative position vector between drone & base station	
a. ΔX = X drone - X base	
b. ΔY = Y drone - Y base	
c. ΔZ = Z drone - Z base	
5. RelativePosition ← CalculateRelativePosition(ECEF_Coordinates, BaseStationCoordinates)	
6. // Convert relative position vector to local ENU coordinates	
1. R=−sinsinlongitudecoscoslongitude0	
−sinsinlatitude∗coscoslogitude−sinsinlatitude∗sinsinlongitudecoscoslatitude	
coscoslatitude∗coscoslongitudecoscoslatitude∗sinsinlongitudesinsinlatitude	
7. E,N,UT=R∗ΔX,ΔY,ΔZT	
8. ENU_Coordinates ← ConvertECEFtoENU(RelativePosition, BaseStationCoordinates)	
9. // Calculate pan and tilt ang	
a. PanAngle ← CalculatePanAngle(ENU_Coordinates)	
b. pan = atan2(E, N)	
c. TiltAngle ← CalculateTiltAngle(ENU_Coordinates)	
d. tilt=atan2U,sqrtE2+N2	
10. // Apply the IMU orientation data	
AdjustedAngles ← ApplyIMUOrientation(PanAngle, TiltAngle, DroneIMUData)	
11. // Control the pan–tilt mechanism	
MovePanTiltController(AdjustedAngles)	
12. return AdjustedAngles	

Data transmission

In this section, data conversions and data transmission successful rate algorithms are presented.

Data conversion

Converting data to laser format is required and this section will show how the data is converted into laser signals. The process of converting various types of data, such as text, images, or sound, into binary data and transmitting them as laser signals involves several steps.

Data conversion to binary.

The data in LCS is text, image and sound. Text data is typically encoded using a character encoding scheme such as ASCII (American Standard Code for Information Interchange) or Unicode. Each character is represented by a unique binary code, allowing the text to be converted into a binary data stream. Images are usually represented as a matrix of pixels, with each pixel having a specific colour value. The colour value can be converted into binary data using various encoding schemes, such as JPEG or PNG. The image data is then serialized into a binary data stream. Sound data is commonly represented as a series of samples of the audio waveform, taken at regular intervals. These samples are then quantized and encoded using an audio codec, such as MP3 or WAV, and converted into binary data. Conversion to binary format is required as shown in the data to binary conversion Algorithm 6 below.

Binary data to laser signals.

Once the data is converted into a binary data stream using the algorithm below, it can be transmitted using the laser communication system. The transmitter microcontroller will be programmed to read the binary data and modulate the laser’s intensity (turn it on and off) according to the binary sequence and following, effectively encoding the data as laser signals. The laser diode is turned on and off (pulsed) according to the binary data. A ‘1’ in the binary data turns the laser on, while a ‘0’ turns it off. This on-off pattern of the laser light encodes the information being transmitted.

Algorithm 6. Data to Binary:	
Data to Binary Conversion Algorithm : Converts textual data into binary code, allowing for laser transmission. This is vital for ensuring the integrity of the data sent over the LCS.	
Input: User text input	
Output: Binary data which can be used to control the laser signals	
1. binary_data = “”	
2. for each character in text do	
3. binary_data + = binary_representation_of(character)	
4. end for	
5. return binary_data	

Laser signals to binary data.

At the receiving end, a solar cell or photodiode is used to detect the laser signals. The received signals are then processed by the receiver microcontroller, which demodulates the laser intensity variations and converts them back into a binary. The receiver microcontroller processes the analogue signal to detect the on-off pattern of the laser light. This is achieved by comparing the received signal to a threshold value. If the signal is above the threshold, it is considered as a ‘1’, and if below the threshold, it is considered a ‘0’ as explained in the Algorithm 7 below.

Binary data to original data.

Finally, the binary data stream is decoded and converted back into its original format of text, images or sound ( Algorithm 8 ). Text: The binary data is interpreted using the same character encoding scheme (e.g., ASCII or Unicode) as in the initial conversion, and the corresponding characters are reconstructed to form the original text. Images: The binary data is deserialized and the original pixel matrix is reconstructed. The image can then be displayed or stored using the same encoding format (e.g., JPEG or PNG) as before. Sound: The binary data is decoded using the same audio codec (e.g., MP3 or WAV) as in the initial conversion, and the audio waveform is reconstructed. The sound can then be played back or stored in its original format. The process of synchronisation is an essential component of every communication system, including laser communication. The program will need to ensure that both appropriate encoding/decoding is carried out and following a synchronization scheme. This would make sure that the transmitter and receiver are in sync with one another, which enables the receiver to appropriately interpret the data that was sent.

Algorithm 7. Laser Signals to Binary:	
Laser Signals to Binary Conversion Algorithm : Translates the laser pulses received by the solar cell back into binary data, ensuring the LCS receiver can reconstruct the original data.	
Input: Laser signals read through solar cell	
Output: Binary data	
1. binary_data = ””	
2. while laser_signal_is_present() do	
3. if laser_intensity() >threshold do	
4. binary_data + = ‘1’	
5. end if	
6. else do	
7. binary_data + = ‘0’	
8. end else	
9. delay()	
10. end while	
11 return binary_data	

The innovation in Algorithms 6 and 7 lies in their efficiency and precision in data conversion. They leverage the unique properties of laser signals to transmit data, offering higher bandwidth and reduced interference compared to traditional RF communication methods. This enables the LCS to transmit data with greater fidelity, crucial for applications requiring high data integrity such as telemedicine.

Algorithm 8. Binary to Data:	
Binary to Original Data Conversion Algorithm : Decodes binary data back into its original form after transmission, essential for the receiver to interpret the data correctly.	
Input: Binary data converted from laser signals	
Output: Transmitted data	
1. text = ””	
2. for each byte in binary_data do	
3. text + = character_representation_of(byte)	
4. end for	
5. return text	
	

Data transmission success rate

The transmission success rate is a critical metric in evaluating the performance of the data transmission system. It is defined as the percentage of successfully transmitted and correctly received data units out of the total data units sent during a given time period or over a given number of trials. In the context of these scenarios, a data unit is considered successfully transmitted if the data received matches the data sent.

For the implementation, each transmission session begins with sending a unique identifier signal to indicate the start of a session. Then, numbers from 1 to 10 are transmitted sequentially. Another unique identifier signal is sent at the end of the transmission session. On the receiver side, upon detecting the start signal, it begins to capture the incoming data until the end signal is detected. After each session, the receiver compares the received data with the expected set of numbers (1 to 10). The transmission success rate is then calculated as the ratio of the correctly received numbers to the total numbers sent (10 in this case), multiplied by 100 to convert it to a percentage. This process is repeated for different conditions in each scenario, and the success rate is recorded for analysis.

In all scenarios, the transmission success rate serves as a quantifiable and comparable measure, allowing to systematically evaluate the prototype’s performance under different conditions and to identify areas for potential improvement. Both the transmitter and receiver code will be modified to only send the specific characters for measuring the transmission rate as shown in Algorithms 9 and 10 .

Algorithm 9. Data Transmission Success Rate for Transmission:	
DTSR Algorithm (Transmission Side) : Assesses the transmission success rate, measuring the accuracy and reliability of the LCS in real-time data transfer scenarios.	
Input: User choice for starting	
Output: Transmit laser signals	
1: define transmitNumber(number) function	
2: convert number to char and store in variable c	
3: transmit c using transmitChar function	
4: transmit newline character using transmitChar function to signal end of a number	
5: end function	
6: define transmitChar(c) function	
7: set ledPin to HIGH (start bit)	
8: delay for transmitDuration microseconds	
9: set ledPin to LOW	
10: for i = 0 to 7 do (for each bit in the character)	
11: set ledPin to the ith bit of c	
12: delay for transmitDuration microseconds	
13: end for	

14: set ledPin to LOW (stop bit)	
15: delay for transmitDuration microseconds	
16: end function	
17: define loop() function	
18: transmit ’#’ character using transmitChar function to signal start of transmission sequence	
19: for i = 1 to 10 do	
20: transmit number i using transmitNumber function	
21: delay for 100 ms	
22: end for	
23: transmit ’$’ character using transmitChar function to signal end of transmission sequence	
24: delay for 1000 ms	
25: end function	

Algorithm 10. Data Transmission Success Rate for Receiver:	
DTSR Algorithm (Receiver Side) : Evaluates the LCS’s data reception fidelity. It captures laser signals, converts them to binary data, and computes the Transmission Success Rate, reflecting the accuracy of the received data against what was transmitted.	
Input: Laser signals received on the solar cell	
Output: DTSR value	
1: define receiveChar() function	
2: delay for transmitDuration milliseconds	
3: initialize receivedByte as 0	
4: for i = 0 to 7 do	
5: if the analog read from SOLARPIN is greater than THRESHOLD then	
6: set the ith bit of receivedByte to 1	
7: end if	
8: delay for transmitDuration milliseconds	
9: end for	
10: delay for transmitDuration milliseconds	
11: return the receivedByte as a char	
12: end function	
13: define loop() function	
14: reading <- analogRead(SOLARPIN)	
15: if reading is greater than THRESHOLD then	
16: receivedChar <- receiveChar() function	
17: if receivedChar is ’#’ then	
18: inTransmissionSession <- true	
19: totalChars <- 0	
20: receivedChars <- 0	
21: else if receivedChar is ’$’ then	
22: inTransmissionSession <- false	
23: calculate successRate as (receivedChars / totalChars * 100)	

24: print “Transmission success rate %”	
25: else if inTransmissionSession is true then	
26: if receivedChar is in range ‘1’ to ‘10’ then	
27: increment totalChars by 1	
28: increment receivedChars by 1	
29: else if receivedChar is in range ‘0’ to ‘10’ then	
30: increment totalChars by 1	
31: end if	
32: end else if	
33: end if	
34: end function	

Results and Discussion

The prototype involves several key components, including the design and development of the prototype system, the implementation of the laser communication module and drone relay mechanism, and the evaluation of the system’s performance under various conditions. These components are critical for demonstrating the system’s feasibility, functionality, and effectiveness and for addressing any concerns or challenges that may arise during the research process. For the prototype to provide sufficient results to support the research, it should demonstrate the following:

Based on our metric the prototype should demonstrate the data transmission success rate (DTSR) successful transmission of data using laser communication. This involves ensuring that data sent from the transmitting station is accurately received at the receiving station. The DSTR will be evaluated based on the conditions listed below:

• Variety of data: The versatility of the prototype will be exhibited by transmitting various types of data. This includes text, images, and sounds, showing its potential for wide-ranging applications.

• Tracking algorithms: The prototype is equipped with tracking algorithms on both the transmitter and receiver. These algorithms are designed to maintain a clear line of sight between the communication points by actively tracking the drone’s movements. The effectiveness of these algorithms will be demonstrated by showing that the system can maintain a stable communication link, despite movements of the drone or the transmitter/receiver.

• Efficiency in varying conditions: To illustrate the robustness and adaptability of the prototype, it will demonstrate efficient data transmission under a variety of conditions. These could include different lighting situations and drone altitudes, simulating real-world scenarios where these factors can fluctuate. The prototype will be tested under these varying conditions to confirm its ability to maintain effective communication links regardless of environmental circumstances.

Data transmission in indoor environment (controlled lighting)

The first scenario considers an indoor environment with controlled lighting conditions in healthcare environment (e.g., homecare). This scenario tests the prototype’s ability to transmit data effectively in a controlled environment without any external interference. The data transmission will be evaluated for various types of data, including text, images, and audio files. The effectiveness of the PAT system in maintaining the direct line of sight between the transmitter and receiver will be assessed. The system’s ability to filter out any ambient light and minimize the impact of indoor lighting will also be assessed. In the test case involving controlled dim lighting conditions, the laser communication system exhibited a stable communication link and reliable data transmission for all three types of data. The system successfully maintained connectivity and transmitted the data without any disruptions or errors.

Similarly, in the test case with controlled bright lighting conditions, such as using a flashlight or simulating sunlight through windows, the laser communication system continued to demonstrate a stable communication link and reliable data transmission. However, it required a minor adjustment in the threshold value to optimize the system’s performance under the specific lighting conditions.

The controlled indoor environment offers a baseline for the prototype’s performance, establishing a transmission success rate under optimal conditions. This serves as a comparison point for subsequent scenarios, offering insights into the impact of varying conditions on the system’s performance. The data collected from Scenario 1 presents a compelling case for the robustness of the laser communication system in an indoor environment under different lighting conditions. For the dim lighting condition, the successful transmission rate was consistently at 100%, indicating a highly reliable communication link despite the lower light levels. However, in bright lighting conditions, the successful transmission rate varied between 90% and 100% as shown in Fig. 4. While still high, this slight variation suggests that brighter lighting conditions could potentially introduce minor disruptions to the laser communication system. Nonetheless, the overall successful transmission rate remained impressively robust across the tested lighting conditions.

With the help of the developed prototype, it may be easily suggested that the laser communication system can work under most types of lighting. The results from this scenario can serve as a benchmark for the other tests mostly in terms of data transmission and help in understanding the system’s capabilities.

Data transmission in indoor environment (varying light)

In the second scenario, the transmission success rate is used to evaluate the system’s resilience under different lighting conditions. By comparing the success rates under dim light, bright light, and fluctuating light intensity to the baseline scenario, we can quantify the effect of lighting disturbances on the system’s ability to maintain reliable communication. Two test cases were conducted to evaluate the performance of the laser communication system under different lighting conditions. The first test case involved gradually increasing and decreasing lighting conditions, while the second test case simulated rapidly changing lighting conditions such as flickering lights and moving shadows.

Figure 4 Lighting conditions vs. transmission rate.

Demonstrates the system’s robustness in maintaining high transmission rates under both dim and bright lighting conditions.

In both cases, the expected outcome was to achieve a stable communication link and reliable data transmission for all three types of data. The laser communication system successfully delivered on these expectations, demonstrating its robustness and ability to maintain a stable connection even in challenging lighting environments. However, it was observed that a minor adjustment in the threshold value was needed to optimize the system’s performance under both scenarios. By fine-tuning the threshold value, the system was able to adapt and ensure efficient data transmission.

The laser communication system was tested under varying lighting conditions in this scenario, specifically dim, bright, and fluctuating lighting. Under dim lighting conditions, the system maintained a perfect transmission rate of 100%, indicating the system’s resilience in low-light environments. During the tests in bright lighting conditions, the successful transmission rate experienced a drop, ranging from 80% to 100%. This suggests that bright lighting might introduce slight disruptions in the laser communication system. The most challenging condition was fluctuating lighting, where the successful transmission rate varied between 60% and 80% as shown in Fig. 5. This indicates that rapid changes in lighting conditions could affect the system’s performance. Nonetheless, the overall transmission rates under all tested conditions suggest the system’s potential for reliable data transmission in varying indoor lighting conditions.

Figure 5 Varied lighting vs. transmission rate.

Illustrates the system’s adaptability to varying light conditions, from dim to bright to fluctuating, ensuring reliable data transmission.

Although in real-world applications, the used laser would be an infrared laser with much higher wavelengths and mostly transparent to the human eye, this scenario test can help verify the effectiveness of the communication system when supported by the drone and PAT system. When a variable threshold value for the laser light is implemented for this prototype system, it could easily detect the variation of the laser’s value from the surrounding lights and was able to differentiate the laser signals.

In emergency healthcare situations, it may be necessary for communication systems to function in poorly lit areas or under fluctuating lighting conditions. This scenario evaluates the system’s ability to maintain reliable data transmission under less-than-ideal lighting conditions, which is essential for ensuring consistent communication during emergencies.

Distances between transmitter and receiver

For this scenario, the transmission success rate becomes a measure of the system’s range capability. By testing at varying distances between the transmitter and receiver, we can determine how distance affects the ability of the system to transmit data effectively. A decrease in the transmission success rate at greater distances would indicate a potential range limitation. The first test case involved testing the system’s ability to track a drone moving in various directions, while the second test case focused on the independent movement of the laser transmitter and receiver endpoints.

In both cases, the actual output showed that the system was able to track the drone successfully. However, it was observed that the system had a slow response time due to the computational speed of the microcontrollers used in the prototype. This slow response time affected the system’s ability to adjust the laser pointing accurately and quickly in response to the drone’s movements.

In Scenario 3, the laser communication system was tested for its performance over varying distances. For short distances, up to 2 m, the system maintained a perfect transmission rate of 100%, demonstrating its robustness for close-proximity applications. However, as the distance increased beyond 2.5 m, the transmission rate started to decline. Between 2.5 and 3 m, the successful transmission rate was 80%. From 3.5 to 4 m, the transmission rate dropped further to 60% and 30% respectively. Finally, beyond 5 m, no successful transmissions were observed. This data suggests that while the system performs well at shorter distances, its effectiveness decreases as the distance between the transmitter and receiver increases as shown in Fig. 6. This insight is essential to understand the range of limitations of the current system and identifying areas for improvement.

Figure 6 Distance vs. transmission rate.

The laser communication system’s successful data transmission rate as the distance increases, highlighting the system’s effective range.

Packet size between transmitter and receiver

In the fourth scenario, the transmission success rate is assessed with varying packet sizes: small (512 bytes), medium (1,024 bytes), and large (2,048 bytes). The evaluation of different packet sizes serves to determine the system’s capacity and reliability when tasked with transmitting different data volumes.

Packet size represents the amount of data the system can transmit in a single operation. In this case, a small packet of 512 bytes might represent a simple text message, while a medium packet of 1,024 bytes could hold more complex data, such as a small image or a lengthy document. A large packet of 2,048 bytes, on the other hand, could contain an audio file or a high-resolution image.

By comparing the success rates for these three packet sizes with the baseline scenario, we can assess how packet size influences the system’s overall performance and robustness. Such an understanding can guide us in configuring the system optimally to meet diverse application needs, enhancing the versatility and adaptability of the laser communication system.

For small packets (512 bytes), the average success rate under dim, bright, and fluctuating light conditions was 100%, 93%, and 89% respectively. This indicates that the system effectively handled smaller data volumes under these diverse lighting situations, albeit with a slight decline in fluctuating light conditions. Medium-sized packets (1,024 bytes) saw average success rates of 97%, 87%, and 81% under dim, bright, and fluctuating light respectively. These figures suggest that while the system can accommodate a higher data volume, the success rate decreases somewhat, particularly under fluctuating light conditions. With large packets (2,048 bytes), the success rates dropped further to 97%, 82%, and 76% under dim, bright, and fluctuating light conditions respectively. While the system managed to transmit large volumes of data, the lighting conditions played a more pronounced role in affecting the transmission success rate.

These results showcase how packet size, in combination with lighting conditions, influences the system’s overall performance as shown in Fig. 7. Despite some decrease in success rates with larger packet sizes and fluctuating light conditions, the system demonstrated a strong ability to adapt to a variety of scenarios. This illustrates its potential for a wide range of applications and sets the stage for future improvements.

Figure 7 Packet size and lighting condition impact on DTSR.

Compares the successful transmission rates for different packet sizes under diverse lighting conditions, showcasing the system’s efficiency.

Furthermore, the study’s results hold particular significance for healthcare, where reliable communication can be a matter of life and death. The system’s high DTSR in controlled and varying light conditions demonstrates its potential to support critical healthcare operations, such as remote patient monitoring and emergency telemedicine, without the risk of RF interference that could affect medical equipment.

The discussion further examines the prototype’s application in healthcare, analysing its operational efficiency and the implications of DTSR findings. The prototype’s success in varied lighting conditions reinforces its suitability for hospital environments, where conditions can change rapidly, ensuring uninterrupted patient care through reliable data transmission.

This innovative laser communication system, enhanced with drone technology, demonstrates notable adaptability in varied environmental conditions and scenarios. However, it is important to acknowledge certain limitations, such as potential range constraints in laser communication and the dependence on drone battery life and stability. Moreover, ethical considerations must be addressed, particularly regarding privacy and data security in healthcare applications. The deployment of drones for communication necessitates stringent measures to safeguard patient data and prevent unauthorized access, ensuring compliance with healthcare regulations and ethical standards. Future developments should focus on enhancing the system’s range capabilities and reinforcing data security protocols to mitigate these challenges effectively.

Conclusion

This study has successfully developed a UAV-assisted indoor laser communication system tailored for healthcare applications. Demonstrating resilience under variable conditions, the system shows promise as a reliable substitute for RF systems that could interfere with medical devices. Key findings indicate its robust performance, notably using infrared lasers, which maintain stable communication links, pivotal in healthcare environments. Future work will extend testing to various conditions and data types, assessing integration with healthcare technology and the system’s capacity for more complex data transfers.

Supplemental Information

Supplemental Information 1 Code for all the microcontrollers and the Raspberry Pi used

Supplemental Information 2 Scenario 1 Dataset

Supplemental Information 3 Scenario 2 Dataset

Supplemental Information 4 Scenario 3 Dataset

Supplemental Information 5 Scenario 4 Dataset

Additional Information and Declarations

Competing Interests

Author Contributions

Data Availability

Tawfik Al-Hadhrami and Faisal Saeed are Academic Editors for PeerJ.

Adeeb Sait conceived and designed the experiments, performed the experiments, performed the computation work, prepared figures and/or tables, authored or reviewed drafts of the article, and approved the final draft.

Tawfik Al-Hadhrami conceived and designed the experiments, performed the experiments, performed the computation work, prepared figures and/or tables, authored or reviewed drafts of the article, and approved the final draft.

Faisal Saeed conceived and designed the experiments, analyzed the data, prepared figures and/or tables, authored or reviewed drafts of the article, and approved the final draft.

Shadi Basurra conceived and designed the experiments, analyzed the data, prepared figures and/or tables, authored or reviewed drafts of the article, and approved the final draft.

Sultan Noman Qasem conceived and designed the experiments, analyzed the data, performed the experiments and the computation works, prepared figures and/or tables, authored or reviewed drafts of the article, and approved the final draft.

The following information was supplied regarding data availability:

The codes are available in the Supplementary File.

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
