# Peer review of "Laser communications system with drones as relay medium for healthcare applications"

_PeerJ Computer Science, doi:10.7717/peerj-cs.1759_

## Round 0.1 · original submission · Minor Revisions

Please see the detailed reviewers' comments. As the reviewers suggested, there's a lack of clarity regarding how the relay drone can identify the transmitter and receiver in dynamic environments. Also, the study should clarify the distinction between real-time communication and executing proposed algorithms. Graphical illustrations should be incorporated to clarify concepts like modulation and transmission protocols. Finally, the results section needs enhancement to demonstrate the relevance of the contribution in the healthcare field.

**Language Note:** PeerJ staff have identified that the English language needs to be improved. When you prepare your next revision, please either (i) have a colleague who is proficient in English and familiar with the subject matter review your manuscript, or (ii) contact a professional editing service to review your manuscript. PeerJ can provide language editing services - you can contact us at copyediting@peerj.com for pricing (be sure to provide your manuscript number and title). – PeerJ Staff

·

Basic reporting

With the promotion of Laser communication and beam communication, the laser applications are rapidly increasing. This is due the attractive for research community. In this paper the authors have provided a very interesting and genuine application for laser communication in health application, and this can be applied in different field. This study has proposed a prototype of a laser communication system-UAV has been designed in real testbeds. This system can transmit data signals (audio and text) using laser technology. The prototype has been designed to control the laser communication system and manage data transmission using UAV. At the same time, the UAV drone was adapted to carry and stabilize the laser communication module during the journey.

Experimental design

Overall, the article seems to be good and interesting but in addition, the following changes are required to be incorporated for this article:
1. The keywords are missing in the paper
2. The abstract are big, so minimize the abstract and make it focus without any more details
3. The introduction section seems a little redundant, the author should focus on the topic of the article description.
4. The algorithm in the proposed work could use some descriptive introduction.
5. What are the outstanding advantages of doing these algorithms, how it effects the communication during transmitting data from sender to the receiver?
6. Titles need more explaining words to be added that will make sense and make it easy for the reader to understand what is the contains for this section.
7. Figure 3 should be readable; fonts are not easy to read.
8. more details about the algorithm 5 PAT system.
9. Algorithms can be added in tables for sender and receiver, if possible, just a recommendation

Validity of the findings

1. The prototype has provided several scenarios with data transmission through drone between sender and receiver and how the reflection has been carried out during the communications.
2. The author should revise the conclusion to strengthen the logical expression;
3. more details are required in the discussion section
4. Reference and are good and I suggest the author add some excellent journal articles from recent years if possible.
Additional comments
In this paper the laser communication has high optimization in data transmission between the sender and receiver through drone reflection. This can be applied in different fields such as Energy and transports, so the contribution is decent and clear.

Additional comments

Excellent work

Reviewer 2 ·

Basic reporting

Abstract should be revised to include brief summary of the research work.

Experimental design

Pl compare the work with an existing recent work.

Validity of the findings

Results should be analyzed in detail.

Additional comments

Introduction should be revised: (1) Introduce the problem (2)discuss about some of the existing research works (3)identify the gap or scope of improvement (4) discuss in order to address the identified gaps what is the methodology used (5) list out the contributions (6) Organization of paper.

I can't find section numbering which totally misguides the reader/reviewer.

Related work should include some solid recent works along with research gaps.

The proposed work should have strong mathematical model.

Write the significance of the Transmitter/Receiver/GPS Algorithms. Also, they need to be written in a structured way.

I can see lots of content which are insignificant (background content). They can be removed to make the work solid.

Conclusion should state the applications of the proposed work.

Reviewer 3 ·

Basic reporting

The paper on laser communications through Uncrewed Aerial Vehicle (UAV) Drones submitted by the author, the paper is well-structured and well-written, however, a number of comments are listed below.
1. The abstract is very long, it should be shortened.
2. recent papers must be cited in the introduction and related work
3. the problem statement is not clear in the related work.
4. The metrics are listed in lines 91 -119, how these are linked to the work in this paper, if these are not linked, then they should be removed.
5. Uregulated spectrum in line 147 is out of the format
6. State the functions of Drone in the structure of this work

Experimental design

The work in this paper seems that it has been implemented in a real testbed which is good work, however, several comments are listed below:
1. The algorithms and pseudocode provided must be explained a bit in-depth, how the communication happens, and what is the Drone job in terms of reflecting the laser path.
2. the algorithms must also have further details in the text, how they work, and affections on the packets and direct them in the correct way.
3. an example of the pointing and tracking algorithm should be presented and explained to support the algorithm and explain why the outcome has resulted in this way.
4. Explain the innovation of algorithms 6 and 7 related to data conversion

Validity of the findings

Results and discussion are well presented and explained however comments such as:
1. the metrics that you have listed and discussed in this section are different from the previous ones in lines 91 -119, explain the relationship between them and how they overlap with each other from a communication point of view.
2. explain more in-depth the scenarios and their conditions
3. the conclusion is very long, shorten the conclusion

·

Basic reporting

The paper presents a well-crafted exploration of a prototype for a laser communication system integrated with a UAV, tested in real-world scenarios. This innovative system enables the transmission of audio and text data using laser technology. The prototype has been meticulously designed to oversee the laser communication system and manage data transmission via the UAV. Concurrently, the UAV drone (DJI Tello) was adapted to carry and stabilize the laser communication module during its operation.

However, there are specific corrections needed before this article can be published:

Abstract Revision: The abstract is lengthy and requires revision to condense the content effectively while retaining its key points.

Missing Keywords: The paper lacks keywords, which are essential for proper indexing. Please provide relevant keywords to enhance the paper's visibility and searchability.

Further Discussion of Limitation: In lines 60-62, the limitation of existing systems is mentioned. Specifically, the constraint related to the direct line of sight requirement for laser communication systems in remote areas needs further elaboration. Additionally, please provide relevant supporting references for this limitation.

Clarity in Research Objective: The section describing the research objective contains mixed aims and adjectives. It is crucial to clearly state the main objective of the research and specify the contributions made. Focus on articulating the primary aim and the unique contributions of this research, especially concerning the development of a reliable, high-speed communication solution for remote and inaccessible areas using laser communication systems and UAV drones as relay media.

Adherence to Journal Template: Please ensure adherence to the journal's template, including the numbering of sections. Numbering the sections will enhance the paper's organization and readability.

Addressing these points will significantly improve the clarity, organization, and overall quality of the paper, making it suitable for publication.

Experimental design

The title of the proposed work needs to be revised and placed within the Methods section.
The following segments should be reformatted as algorithms, namely:
- Algorithm 4: Sending GPS values to Laser Pointer from the drone (Piont 358)
- Algorithm 3: Moving the Laser Pointer towards the drone (Piont 357)
- Algorithm 1: Laser Transmitter for LCS (Point 345)
and any other relevant sections.
It would be beneficial to summarize or group these algorithms within a figure for better visualization and understanding.

Validity of the findings

The results have been thoroughly examined and presented in detail. The proposed solution underwent evaluation based on transmission success rates, with various scenarios explored. Nonetheless, it is essential to emphasize the limitations of prior research efforts and recommend directions for future work.

Additional comments

The part of conclusion highlights the successful development of an indoor laser communication system using drones for healthcare applications. The system demonstrated adaptability but lacked discussion on limitations and ethical considerations. Future research directions were mentioned, but there was no detailed comparative analysis with existing technologies.

---

## Round 0.2 · accepted · Accept

Reviewers have confirmed the authors have addressed all of their comments.

Reviewer 3 ·

Basic reporting

The revised versions have been corrected based on my previous comments. The manuscript could be accepted in the current form.

Experimental design

The revised versions have been corrected based on my previous comments. The manuscript could be accepted in the current form.

Validity of the findings

The revised versions have been corrected based on my previous comments. The manuscript could be accepted in the current form.

Additional comments

The revised versions have been corrected based on my previous comments. The manuscript could be accepted in the current form.

·

Basic reporting

Corrections are as we previously mentioned. It is accepted and satisfied.

Experimental design

Corrections are as we previously mentioned. It is accepted and satisfied.

Validity of the findings

Corrections are as we previously mentioned. It is accepted and satisfied.

Additional comments

Corrections are as we previously mentioned. It is accepted and satisfied.